

# Does livestock ownership predict animal-source food consumption frequency among children aged 6–24 months and their mothers in the rural Dale district, southern Ethiopia?

Tsigereda Kebede[1,2], Selamawit Mengesha Bilal[2,3], Bernt Lindtjorn[1,2] and Ingunn M. S. Engebretsen[1]

[1] Centre for International Health, University of Bergen, Bergen, Norway
[2] College of Medicine and Health Sciences, Hawassa University, Hawassa, Ethiopia
[3] Sidama Regional State Health Bureau, Hawassa, Sidama Region, Ethiopia

## ABSTRACT

**Background:** Animal-source foods are food items that come from animals. Animal-source foods provide a variety of micronutrients that plant-source foods cannot provide to the same extent and without extra precaution. Milk, eggs, poultry, flesh meat and fish are animal-source foods mainly used in Ethiopia.

Low animal-source food consumption among children and mothers is a great concern in many low-income settings. This study aimed to describe animal-source food consumption frequencies among children aged 6–24 months and their mothers in rural southern Ethiopia where livestock farming is very common. We also analysed the association between livestock ownership and animal-source food consumption among children and mothers.

**Methods:** A community-based cross-sectional study was conducted among 851 randomly selected households with child-mother pairs from August to November 2018. The study was conducted in the rural Dale District, southern Ethiopia. Structured and pre-tested questionnaires were used to collect data on mother and child information, livestock ownership, and animal-source foods consumption frequencies. Ordinal logistic regression analysis was used to describe associations between animal-source foods consumption and livestock ownership.

**Result:** Nearly, three-quarters (74.1%) of the households owned cows, and a quarter (25%) had goats or sheep. Dairy, egg and meat consumption among children during the past month was 91.8%, 83.0% and 26.2%, respectively. Likewise, the consumption of dairy, eggs and meat among mothers was 96.0%, 49.5% and 34.0%, respectively. The percentage of children who had not consumed any animal-source foods during the month prior to our survey was 6.6%, and the figure was 2.2% for the mothers. Dairy consumption was 1.8 times higher among children (aOR = 1.8, 95% CI [1.3–2.5]) and 3.0 times higher among mothers (aOR = 3.0, 95% CI [2.2–4.2]) in households that kept cows than in households without cows. The egg consumption frequency was positively associated with hen and goat/sheep ownership for both children and mothers. Meat consumption frequency among children was negatively associated with cow ownership (aOR = 0.66, 95% CI [0.45–0.95]); however, cow ownership was not associated with meat consumption among mothers.

Corresponding author
Tsigereda Kebede,
tsige_behailu@yahoo.com

**Conclusion:** Dairy products were common animal-source foods consumed by young children and mothers in the study area. However, meat consumption was low among children and mothers. Strategies like promoting the keeping of goats/sheep and hens to improve complementary feeding and mothers' nutrition are warranted in the study area.

# INTRODUCTION

Child and maternal nutrition plays a crucial role in the overall health and well-being of individuals, families, and communities. It is widely recognized that proper nutrition during pregnancy and early childhood is essential for optimal growth, development, and lifelong health outcomes (*Koletzko et al., 2019*). Child and maternal undernutrition is a global concern associated with high rates of morbidity and mortality (*Prendergast & Humphrey, 2014*). Nutritional deficiencies during pregnancy and early childhood cause negative pregnancy outcomes and sub-optimal child growth. Insufficient intake of both macro- and micro-nutrients can have profound effects on child and maternal health outcomes (*Elmadfa & Meyer, 2017*). In resource-poor settings where nutrient intake is inadequate, the lack of animal-source foods (ASFs) in diets contributes to child and maternal undernutrition (*Patel et al., 2018*; *Victora et al., 2022*). Ethiopia has significant challenges related to child and maternal undernutrition including low ASFs consumption (*Central Statistical Agency (CSA) (Ethiopia) and ICF, 2016*; *Ethiopian Public Health Institute E and ICF, 2019*).

Animal-source foods are rich sources of high-quality nutrients essential for normal reproductive function and optimal child growth (*Adesogan et al., 2020*; *Iannotti et al., 2017*). Milk, for instance, provides energy, protein, and micronutrients that nurture the young, and stimulate growth (*Dror & Allen, 2011*; *Herber & Bogler, 2020*). Eggs contain key nutrients that enhance foetal growth during pregnancy, improve breast milk composition during lactation and support child growth in early childhood (*Lutter, Iannotti & Stewart, 2018*). Meat is a valuable source of protein, iron and vitamin B12, which are either absent or poorly absorbed from other sources (*McAfee et al., 2010*). Fish and poultry are also important sources of protein and micronutrients (*Alagawany et al., 2018*; *Balami, Sharma & Karn, 2019*). ASFs are also important sources to provide essential Amino Acids which cannot be synthesized by humans, such as lysine, methionine, and tryptophan. Different ASFs differ in their fat content, with fish being lower in fat content compared to other ASFs. Therefore, promotion of diverse ASFs in diets is critical for addressing child and maternal undernutrition globally. There is a global dilemma concerning whether meat consumption could sustainably increase among low-meat eaters for health purposes. Most argue that there should be a reduction among high-meat consumers and that consumption should ideally be more equitable (*Bimbo, 2023*; *Wertheim-Heck & Raneri, 2019*).

Livestock production is the largest component of the Ethiopian agricultural sector, serving as the backbone of the country's economy (*Gashaw, Asresie & Haylom, 2014*). In fact Ethiopia is recognized as the leading producer and exporter of livestock in Africa. Despite this, consumption of ASFs remains low in most of the Ethiopian population (*Abegaz, Hassen & Minten, 2018*; *Leta & Mesele, 2014*; *Tilahun & Schmidt, 2012*; *Tiruneh et al., 2021*). Moreover, it should be noted that there are considerable differences in ASFs consumption rates across regions within Ethiopia (*Central Statistical Agency (CSA) (Ethiopia) and ICF, 2016*). There are significant variations in ASF consumption among children, with the highest rate (41.7%) being reported in Addis Ababa and the lowest rate (5.9%) reported in the Somalia region. Similarly, ASFs consumption is reported to be low among mothers in the reproductive age group and at household level in Ethiopia (*Bosha et al., 2019*; *Daba et al., 2021*). In certain areas like northern Ethiopia's Tigray regional state, ASFs are primarily consumed on ceremonial occasions, while home-reared livestock and their products are mainly used for income generation purposes (*Haileselassie et al., 2020*).

Studies have found low consumption of ASFs in the general population particularly among children and mothers. For instance, a national survey revealed that only 17.5% of children consumed eggs with in a 24-h period, while 8.7% consumed flesh products (*Hamza et al., 2022*). A study found that milk was consumed by 48% of children, while eggs and meat were consumed by 27% and 11% respectively (*Potts, Mulugeta & Bazzano, 2019*). Another study conducted in the Amhara region of Ethiopia reported an ASF consumption rate of 35.4% among adults (*Keflie et al., 2018*). Despite these findings, there remains a lack of comprehensive research on how livestock ownership impacts ASFs consumption in rural settings. Additionally, there is limited information available in current ASF consumption rates among specific groups such as children aged 6–24 months and mothers. The objectives of this study were: to describe the frequency of ASF consumption among children aged 6–24 months and their mothers, and to investigate associations between livestock ownership and ASFs consumption frequencies. The aim was to thereby improve the understanding of ASFs consumption frequencies in rural Ethiopia.

# METHODS AND MATERIALS

## Ethical statement

Ethical approval was obtained from the Hawassa University College of Medicine and Health Sciences Institutional review board, Ethiopia (reference number; IRB/025/10), and from the Norwegian Regional Ethical Committee (REK), Norway (reference number; 2018/90/REK Vest). All the necessary official letters were obtained from the concerned bodies at regional and district levels. The respondent's signature or finger stamp was obtained before enrolment to signify their willingness to participate in our study. The data has been kept confidential and no personal or household identifiers were used to communicate our findings.

## Study setting

This study was conducted in the rural district, in the Sidama regional state of Ethiopia. The Dale district, comprised of 36 rural and two urban kebeles, is one of the 37 districts (previously it was one of the 19 districts) according to the *Sidama Region Health Bureau (2022)*. A kebele is the smallest administrative unit. This study was conducted in seven rural kebeles of the district. The projected population size of the district in 2022 was reported over 350,000 (*Sidama Region Health Bureau, 2022*). The main town in the district, Yirga Alem, lies 320 km from Addis Ababa. The Dale district has 33 health posts, ten health centres and a hospital called Yirga Alem General Hospital. The people living in the study area are mainly farmers, cultivating maize, ensete (*Ensete ventricosum*), kale, cabbage, and haricot beans. Coffee, chat (*Catha edulis*) and different seasonal fruits are grown in the area. There is also a cultural value of keeping livestock such as cows, goats and sheep. Most rural households also have hens.

## Study design

A community-based cross-sectional study was conducted to assess animal-source food consumption frequencies, livestock ownership and background information. The study was conducted from August to November 2018.

## Study participants

This study included a sample of 851 children aged 6–24 months and caregivers (primarily the mother) out of 971 children aged 0–24 months and caregivers enrolled in a larger study (Household food security and dietary practices in the rural Dale district, southern Ethiopia). Study procedures, including the original sample size estimation, sampling procedures and recruitment are described in a previous article (*Behailu et al., 2022*). A census, household listing, proportionality according to kebele size, and simple random sampling were performed. This study excluded those younger than 6 months, as exclusive breastfeeding is recommended for that age group, so that 853 children aged 6–24 months remained. Of those, two children had no mother or caregiver eligible for dietary data collection (Fig. 1).

## Study variables

The outcome variables in our study were ASFs consumption frequency among children and mothers. ASFs assessed in this study were dairy products (milk, yoghurt, cheese and whey), eggs and meat (flesh meat, organ meat, poultry and fish). Consumption frequency was defined as the usual number of servings per day, week, month and 3 months, irrespective of the portion size served to children and their mothers. Frequencies were stated as 'never' if not consumed in the past 3 months, 'rarely' if consumed in the past 3 months, but not consumed in the past 1 month, 'sometimes' if consumed one to three times in the month prior to this study, 'often' if consumed one to six times in the week prior to this study, and 'always' if consumed at least once daily. For analysis purposes, the

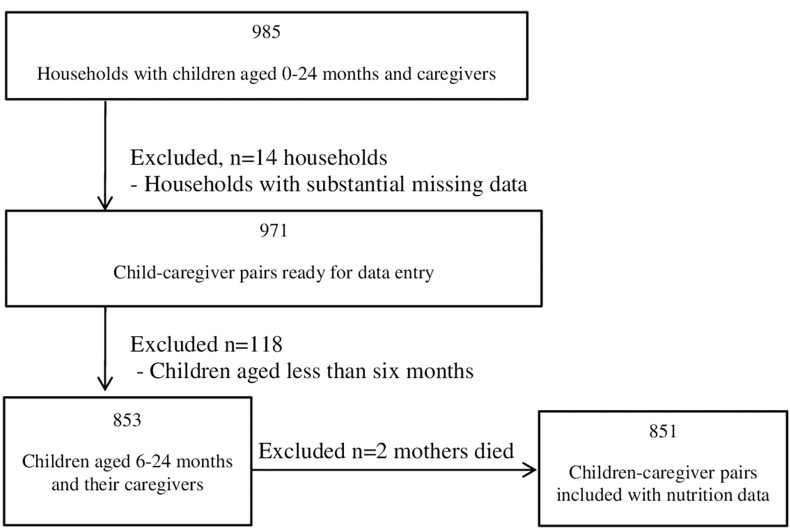

**Figure 1 Study profile that presents children and mothers excluded from the study and the reasons they were excluded.**

frequencies were assigned with ordered numbers: '0' for 'never' and 'rarely', '1' for 'sometimes', '2' for 'often', and '3' for 'always'.

The independent variables were livestock ownership such as cows, donkeys, oxen, goats or sheep, and hens at household level. Child characteristics included age and gender, information about mothers comprised age and educational status, and household background described household size, and food insecurity status.

## Data collection

Data collection procedures and data quality measures are described in our previous article (*Kebede et al., 2022*), explicitly about methods applied to household food insecurity and wealth index analysis.

## Statistical analysis

The data was analyzed using SPSS version 25 (SPSS Inc., Chicago, IL, USA). Frequencies, percentages and means were used to describe continuous and categorical variables. Ordinal logistic regression (OLR) analysis was used to explain the association between the outcome variable, which was a composition variable including dairy, egg and any meat consumption, and predictor variables. Before running the multivariable ordinal logistic regression, bivariable ordinal logistic regression was conducted. Variables with a *P* value < 0.2 in the bivariable ordinal regression were included in the multivariable ordinal logistic regression. Multi-collinearity was also checked to rule out any association between the independent variables before running the adjusted ordinal logistic regression. The unadjusted odds ratio (uOR) and the adjusted odds ratio (aOR), with the 95% confidence intervals (CI), were used to explain associations between the outcome and the exposure variables. *P* value < 0.05 was used to determine the level of statistical significance.

## RESULTS

### Background characteristics

In this study, a total of 851 child-mother pairs were analyzed. Among the children, 50.9% (433) were females and the mean age in months was 15.4 (95% CI [15.1–15.8]). The mean age of the mothers was 26.9 years (95% CI [26.5–27.2]). About half of the mothers, 45.1% (384) had attended primary school. The mean household size was 4.8 persons (95% CI [4.7–4.9]), with 51.5% (438) of households having less than five members. Approximately 54% (459) of the households experienced moderate to severe food insecurity (Table 1). When it comes to livestock ownership, 74.1% (631) of the households owned cows, while 25% (213) owned goats or sheep. Around one-fifth, 20.7% (176) of the households kept all types of livestock including sheep/goats, cows, and hens, whereas 17.4% (148) did not have any of these animals. Oxen and donkeys were found in only a few households; specifically, oxen were kept by 2.9% (25) of the households and donkeys by 6.5% (55) of the households (Table 2).

### Animal-source food consumption frequencies among children aged 6–24 months

More than 85% (745) of the children had consumed dairy products daily or up to six times per week, and 70 children had never consumed dairy products during the past month. Eggs were served one to six times per week for 36.5% (311) of the children, while 17% (146) of the children had not eaten eat eggs during the past month. No child consumed meat on a daily basis. Almost three-fourths, 73.8% (628), had not eaten meat in the past month. Out of the 851 children in total, 6.6% (56) had not consumed any ASFs during the past month (Table 3). In addition to this, 50 (28 female and 22 male) children aged 6 to 9 months were not getting any food other than breast milk. The distribution of ASF consumption frequency among children by each livestock of ownership was provided in Table S1.

### Animal-source food consumption frequencies among mothers

The dairy product consumption of mothers during the past month was 96% (817) of the 851 mothers (Table 4). More than 50% (430) of mothers had not eaten eggs during the past month and 31.3% (266) of the mothers had eaten eggs one to three times per month. Overall, 2.2% (19) of them did not consume any ASFs during the month prior to our survey (Table 4). The distribution of ASF consumption frequency among mothers by each livestock of ownership was provided in Table S2.

### Comparison of animal-source food consumption between children (male and female) and mothers

Animal-source food consumption among children and mothers was varied. While meat and egg consumption was highest among children, dairy product consumption was highest among mothers. ASFs were almost equally consumed among male and female children aged 6–24 months (Table S3).

**Table 1 Socio-demographic characteristics of households, children and mothers in the Dale district, southern Ethiopia (N = 851).**

| Variables | Number (N) | Percentages (%) |
|---|---|---|
| Household size | | |
| Less than five | 438 | 51.5 |
| Five and above | 413 | 48.5 |
| Household food insecurity | | |
| Food secure | 180 | 21.2 |
| Mildly food insecure | 212 | 24.9 |
| Moderately food insecure | 333 | 39.1 |
| Severely food insecure | 126 | 14.8 |
| Households' socioeconomic status | | |
| Lower | 258 | 30.3 |
| Middle | 310 | 36.4 |
| Upper | 383 | 33.3 |
| Child's age | | |
| 6–12 months | 261 | 30.7 |
| 12–18 months | 285 | 33.5 |
| 18–24 months | 305 | 35.8 |
| Child's sex | | |
| Female | 433 | 50.9 |
| Male | 418 | 49.1 |
| Mother's age | | |
| 15–24 years | 263 | 30.9 |
| 25–34 years | 505 | 59.3 |
| 35–49 years | 83 | 9.8 |
| Mother's educational status | | |
| No education | 139 | 16.3 |
| Primary school | 384 | 45.1 |
| Secondary school and above | 328 | 38.5 |

## Animal-source food consumption among child-mother pairs at household level

Animal source food consumption at household level was also described according to child-mother pairs. Households in which only the child, only the mother or both, consumed or did not consume ASFs during the month prior to this study were described. In 13% (111) of the households both the child and the mother had consumed meat, while this was 47% (400) for eggs and 89% (757) for dairy products. Children ate meat and eggs a little more frequently than their mothers, while mothers ate dairy products slightly more frequently than their children (Table S4).

**Table 2 Livestock ownership among rural households in the Dale district, southern Ethiopia (N = 851).**

| Type of livestock owned | Yes | No |
|---|---|---|
| | Number (%) | Number (%) |
| Goats/sheep | 213 (25.0) | 638 (75.0) |
| Cows | 631 (74.1) | 220 (25.9) |
| Hens | 507 (59.6) | 344 (40.4) |
| Donkeys | 55 (6.5) | 796 (93.5) |
| Oxen | 25 (2.9) | 826 (97.1) |
| All types of livestock and hens | 176 (20.7) | 675 (79.3) |
| At least one type of livestock or hens | 703 (82.6) | 148 (17.4) |

Note:
Goats and sheep are combined as 'goats/sheep' as they were counted together in most situations.

**Table 3 Animal-source food consumption frequency among the children during the month prior to this study, Dale district, southern Ethiopia (N = 851).**

| Animal-source foods | Once or more times/ day N (%) | 1–6 times/ week N (%) | 1–3 times/ month N (%) | Did not consume during the past month N (%) | Total N (%) |
|---|---|---|---|---|---|
| Dairy products | 377 (44.3) | 366 (43.0) | 38 (4.5) | 70 (8.2) | 851 (100) |
| Eggs | 91 (10.7) | 311 (36.5) | 304 (35.7) | 145 (17.0) | 851 (100) |
| Meat (flesh meat, organ meat, poultry and fish) | 0 | 40 (4.7) | 183 (21.5) | 628 (73.8) | 851 (100) |
| Any animal-source foods combined | 396 (46.5) | 378 (44.4) | 21 (2.5) | 56 (6.6) | 851 (100) |

**Table 4 Animal-source food consumption frequencies among the mothers during the month prior to this study, Dale district, southern Ethiopia (N = 851).**

| Animal-source foods | Once or more times/day N (%) | One to six times/ week N (%) | One to three times/ month N (%) | Did not consume during the past month N (%) | Total N (%) |
|---|---|---|---|---|---|
| Dairy products | 245 (28.8) | 383 (45.0) | 189 (22.2) | 34 (4.0) | 851 (100) |
| Eggs | 24 (2.8) | 131 (15.4) | 266 (31.3) | 430 (50.5) | 851 (100) |
| Meat (flesh meat, organ meat, poultry and fish) | 0 | 92 (10.8) | 197 (23.1) | 562 (66.0) | 851 (100) |
| Any animal-source foods | 254 (29.9) | 417 (49.0) | 161 (18.9) | 19 (2.2) | 851 (100) |

## Associations between livestock ownership and animal-source food consumption frequencies among children aged 6–24 months

The frequency of dairy products consumption was 1.8 times higher among children in households owning cows than among those which did not (aOR = 1.8; 95% CI [1.3–2.5]). In contrast, the frequency of meat consumption was lower among children in households keeping cows (aOR = 0.66; 95% CI [0.45–0.95]), compared with those that did not have cows. Goat or sheep ownership was associated with a 1.7 times greater frequency of egg

**Table 5 Ordinal logistic regression of animal-source food consumption frequency among children aged 6–24 months by livestock ownership in the Dale district, southern Ethiopia (N = 851).**

| Predictors | Dairy (milk, yoghurt, cheese or whey) | | Eggs | | Any meat (flesh meat, organ meat, poultry or fish) | |
|---|---|---|---|---|---|---|
| | COR (95% CI) | AOR (95% CI) | COR (95% CI) | AOR (95% CI) | COR (95% CI) | AOR (95% CI) |
| **Goats/sheep** | | | | | | |
| No | 1 | 1 | 1 | 1 | 1 | |
| Yes | 3.1 [2.3–4.3] | 2.3 [1.6–3.3][a] | 2.6 [1.9–3.5] | 1.7 [1.2–2.3][b] | 1.2 [0.8–1.7] | |
| **Cows** | | | | | | |
| No | 1 | 1 | 1 | 1 | 1 | 1 |
| Yes | 2.6 [2.0–3.5] | 1.8 [1.3–2.5][a] | 1.4 [1.1–1.9] | 0.8 [0.6–1.1] | 0.8 [0.5–1.1] | 0.66 [0.45–0.95][c] |
| **Hens** | | | | | | |
| No | 1 | 1 | 1 | 1 | 1 | 1 |
| Yes | 2.2 [1.7–2.8] | 1.3 [0.9–1.7] | 3.9 [2.95–5.03] | 3.5 [2.6–4.8][a] | 1.2 [0.9–1.7] | 1.36 [0.97–1.9] |
| **Household Food Insecurity score** | 0.97 [0.9–1.0] | 0.98 [0.9–1.02] | 1.0 [0.98–1.06] | | 0.97 [0.9–1.0] | 0.96 [0.92–1.0] |
| **Child's age** | 1.05 [1.03–1.08] | 1.05 [1.03–1.08][a] | 1.07 [1.04–1.1] | 1.07 [1.04–1.09][a] | 1.06 [1.03–1.09] | 1.06 [1.03–1.09][a] |
| **Mother's age** | 0.98 [0.95–1.0] | 1.0 [0.99–1.06] | 1.0 [0.97–1.03] | | 0.97 [0.9–1.0] | 0.98 [0.95–1.0] |
| **Household size** | 0.83 [0.8–0.9] | 0.84 [0.8–0.9][b] | 0.9 [0.8–0.94] | 0.9 [0.85–1.0] | 1.0 [0.9–1.1] | |
| **Mother's education** | | | | | | |
| No schooling | 1 | 1 | 1 | 1 | 1 | |
| Primary school | 1.8 [1.3–2.6] | 1.3 [0.9–1.9] | 1.7 [1.2–2.3] | 1.2 [0.8–1.8] | 0.9 [0.6–1.5] | |
| Secondary school | 3.1 [2.2–4.1] | 1.9 [1.3–3.0][b] | 1.8 [1.2–2.5] | 1.2 [0.8–1.8] | 1.3 [0.8–1.9] | |

Notes:
[a] $P < 0.001$.
[b] $P < 0.01$.
[c] $P < 0.05$.
All variables with a $P$ value < 0.2 in the bivariable regression are included in the multivariable regression. Household food insecurity (HFI) scores were excluded from the multivariate analysis of egg consumption frequencies. Ownership of goats/sheep, household size, and mother's education were also excluded from the multivariate analysis of meat consumption frequencies due to the $P$ value > 0.2 in the bivariable regression.

consumption among children, than for households without goats or sheep (aOR = 1.7; 95% CI [1.2–2.3]). Furthermore, owning goats or sheep increased the child's dairy product consumption frequency more than twofold (aOR = 2.3; 95% CI [1.6–3.3]). Egg consumption frequency among children was 3.5 times higher among households that kept hens than among those which did not (aOR = 3.5; 95% CI [2.6–4.8]) (Table 5).

## Associations between livestock ownership and animal-source food consumption frequencies among mothers

The ownership of livestock, specifically goats or sheep and hens, was associated with higher frequencies of dairy, egg and meat consumption frequency among mothers. Cow ownership was specifically associated with increased dairy products consumption frequency. While goat or sheep ownership was associated with all ASFs consumption frequency, cow ownership was solely associated with dairy product consumption frequency (aOR = 3.0; 95% CI [2.2–4.2]). The frequency of dairy product consumption among mothers was also greater in households keeping goats or sheep (aOR = 1.9; 95% CI [1.3–2.6]) than among mothers in households without these animals. Regarding egg
**Table 6 Ordinal logistic regression of animal-source food consumption frequency among mothers of children aged 6–24 months by livestock ownership, the Dale district, southern Ethiopia, (N = 851).**

| Predictors | Dairy (milk, yoghurt, cheese or whey) | | Eggs | | Any meat (flesh meat, organ meat, poultry or fish) | |
|---|---|---|---|---|---|---|
| | COR (95% CI) | AOR (95% CI) | COR (95% CI) | AOR (95% CI) | COR (95% CI) | AOR (95% CI) |
| **Goats/sheep** | | | | | | |
| No | 1 | 1 | 1 | 1 | 1 | 1 |
| Yes | 2.6 [1.9–3.5] | 1.9 [1.3–2.6][a] | 3.3 [2.4–4.4] | 2.3 [1.6–3.1][a] | 2.4 [1.8–3.3] | 1.8 [1.3–2.5][b] |
| **Cows** | | | | | | |
| No | 1 | 1 | 1 | 1 | 1 | 1 |
| Yes | 3.6 [2.7–4.9] | 3.0 [2.2–4.2][a] | 2.1 [1.5–2.8] | 1.4 [0.99–1.99] | 2.2 [1.4–2.9] | 1.4 [0.95–2.0] |
| **Hens** | | | | | | |
| No | 1 | 1 | 1 | 1 | 1 | 1 |
| Yes | 1.9 [1.4–2.4] | 1 [0.8–1.4] | 2.9 [2.2–3.8] | 2 [1.5–2.7][a] | 2.2 [1.6–3.0] | 1.5 [1.1–2.2][c] |
| **HFI score** | 0.99 [0.96–1.04] | | 0.8 [0.7–0.9] | 1.06 [1.02–1.11][b] | 1 [0.96–1.05] | |
| **Child's age** | 0.98 [0.96–1.0] | | 1.04 [1.02–1.07] | 1.04 [1.01–1.06][b] | 1.05 [1.02–1.07] | 1.05 [1.02–1.07][b] |
| **Mother's age** | 0.99 [0.96–1.0] | | 1 [0.96–1.01] | | 1 [0.98–1.03] | |
| **Household size** | 0.8 [0.75–0.87] | 0.85 [0.8–0.9][a] | 1.05 [1.01–1.09] | 0.8 [0.67–0.85][a] | 0.9 [0.8–0.96] | 0.95 [0.86–1.05] |
| **Mother's education** | | | | | | |
| No schooling | 1 | 1 | 1 | 1 | 1 | 1 |
| Primary school | 2.4 [0.7–3.4] | 1.6 [1.1–2.3][c] | 2.6 [1.8–4.0] | 1.8 [1.1–2.7][c] | 2.3 [1.4–3.6] | 1.7 [1.1–2.8][c] |
| Secondary school | 3.9 [2.7–5.6] | 2.1 [1.4–3.2][a] | 2.6 [1.7–4.0] | 1.4 [0.9–2.3] | 2.7 [1.7–4.3] | 1.8 [1.1–3.1][c] |

**Notes:**
[a] $P < 0.001$.
[b] $P < 0.01$.
[c] $P < 0.05$.
All variables with $P$ value $< 0.2$ in the bivariable regression are included in the multivariable regression. Mother's age was excluded from all animal-source food consumption frequencies. The household food insecurity (HFI) score was excluded from dairy and meat consumption frequencies, and child's age was excluded from dairy consumption frequencies due to $P$ value $> 0.2$ in the bivariable regression.

consumption, there was a 2.3 and 2.0 greater frequency among mothers in households with goats or sheep (aOR = 2.3; 95% CI [1.6–3.1]) and hens (aOR = 2; 95% CI [1.5–2.7]), respectively. Having hens was also associated with meat and egg consumption frequencies among mothers. Meat consumption frequency among mothers was 1.8 times (aOR = 1.8; 95% CI [1.3–2.5]) higher in households that had goats or sheep than in those that did not. Likewise, ownership of hens was associated with a 1.5 increased frequency of meat consumption among mothers (aOR = 1.5; 95% CI [1.1–2.2]) (Table 6). In general, these findings suggest that having access to livestock may contribute to a higher intake of ASFs in rural households.

## Socio demographic variables associated with animal-source food consumption among children and mothers

Child age, household size and educational status of the mother were associated with ASFs consumption frequencies among children and mothers. Child age was positively associated with dairy (aOR = 1.05; 95% CI [1.03–1.08]), egg (aOR = 1.07; 95% CI [1.04–1.09]) and meat (aOR = 1.06; 95% CI [1.03–1.09]) consumption among children. It was also positively

associated with egg (aOR = 1.04; 95% CI [1.01–1.06]) and meat (aOR = 1.05; 95% CI [1.02–1.07]) consumption among mothers. Secondary school attendance of the mother was positively associated with meat consumption among children (aOR = 1.9; 95% CI [1.3–3.0]), and with dairy (aOR = 2.1; 95% CI [1.4–3.2]) and meat (aOR = 1.8; 95% CI [1.1–3.1]) consumption among mothers. In addition, primary education of the mother was positively associated with all animal-source food consumption among mothers. However, household size was negatively associated with dairy consumption among children (aOR = 0.84; 95% CI [0.8–0.9]) (Table 5), and with dairy (aOR = 0.85; 95% CI [0.8–0.9]) and egg (aOR = 0.8; 95% CI [0.67–0.85]) consumption among mothers (Table 6).

## DISCUSSION

This study showed that ASFs consumption among children and their mothers varied. One fifth never consumed any ASFs, and dairy products constituted the most common ASF. Children had a higher consumption frequency of eggs and meat than their mothers. This study also showed that livestock keeping, especially cows and hens, is common in rural households. Goats and sheep are found among a quarter of the total number of households. While goat/sheep and hen ownership was positively associated with meat consumption, cow ownership was inversely associated with meat consumption in children.

Cow ownership was positively associated with children's milk consumption; the milk consumption frequency was nearly double in households owning cows compared to those that did not. This finding is in line with studies that reported similar positive associations (*Broaddus-Shea et al., 2020*; *Hetherington et al., 2017*; *Temesgen, Yeneabat & Teshome, 2018*). However, cow ownership was not associated with egg consumption and was negatively associated with meat consumption among children. It is clear that cows are not commonly used as a source of meat, rather they keep them as a measure of wealth (*Acharya, Yang & Jones, 2021*), and as a source of income by selling dairy products and calves. However, traditional animal husbandry and low productivity (*Jembere, Kabthymer & Deribew, 2020*) keep households at lower economic status and unable to afford eggs and meat. The current situation in Ethiopia also shows that the cost of ASFs is increasingly high (*Daba et al., 2021*; *Haileselassie et al., 2020*).

Owning goats or sheep and hens demonstrated a positive association with egg consumption among children. The frequency of egg consumption among children from households keeping hens was 3.5 times higher than that of children from households without hens. This positive association has been reported by various studies (*Broaddus-Shea et al., 2020*; *Mosites et al., 2017*). However, there are studies that have found no association between hen ownership and egg consumption (*Dumas et al., 2018*). Geographical and cultural differences between Ethiopia and Zimbabwe may account for this discrepancy. However, the finding of this study suggests that egg consumption among children is positively influenced directly by keeping hens in the household. The association between egg consumption frequency and goats/sheep ownership can be explained in different ways. Having goat/sheep means that the household has a better economic status which enables to afford ASFs including eggs. In this study, we have seen that 90% (191 out

of 213) households keeping goat/sheep also keep hens which can be also the reasons for the association.

Our finding that there was no association between goat/sheep ownership and meat consumption among children is supported by a study from Nepal (*Broaddus-Shea et al., 2020*). The possible explanation for this may be that these animals are primarily used for income generation purposes as reported in different studies (*Bundala et al., 2020*; *Kocho, 2007*; *Tesfa et al., 2021*). A study from Kenya also documents that livestock production is mainly used to generate income for poor rural households in Sub-Saharan Africa (*Jin & Iannotti, 2014*). The high cost of live animals may contribute to low meat consumption at the household level; households do not slaughter animals for household consumption. The current cost of meat is also unaffordable for most households. Additionally keeping hens may be more focused on egg production rather than poultry consumption at household level (*Daba et al., 2021*). It is also noted that feeding poultry to children may not be culturally accepted in some regions.

The higher milk consumption demonstrated in this study may suggest that caregivers consider milk to be enough for young children and ignore other ASFs, particularly meat. However, it is important to note that while milk is good source of calcium and Vitamin D, meat is rich in iron, zinc, and vitamin B12. These micronutrients are crucial for the growth and development of young children (*Dror & Allen, 2011*). In addition, food taboos concerning meat and poultry consumption by children might be a challenge (*Melesse & van den Berg, 2021*). Although no food taboos were reported in this study, other studies in Ethiopia reported social norms and beliefs as common barriers to ASF consumption among young children and mothers (*Haileselassie et al., 2020*). Even though the majority of the households keep hens, the prevalence of childhood stunting was high (39.5%) in our recent publication about the nutritional status of the children in the area (*Behailu et al., 2022*). This finding may suggest that households mainly keep hens for the production of eggs for sale, rather than egg consumption at household level (*Daba et al., 2021*). This gap needs to be strongly addressed and nutrition education, especially about complementary feeding, should be promoted.

Meat consumption among mothers in this study was lower than in a study from Gondar town, North-western Ethiopia (*Aserese et al., 2020*). Cultural variation in dietary practices among mothers, especially after delivery, may explain the difference. The study design can also provide an additional explanation: this was a community-based study, while that in Gondar was a facility-based. On the other hand, meat consumption among mothers in this study was higher than in a study from the Afar region, in which this was reported at 11% (*Mulaw, Feleke & Mare, 2021*). Goat/sheep demonstrated a positive association with meat consumption among mothers. A study from Kenya has reported a similar association (*Thumbi et al., 2015*). However, our study did not assess the number of livestock of each type. Likewise, hen ownership was associated with meat consumption among mothers. Even though eggs were mainly given to children (83%), around 50% of the mothers had consumed eggs during the past month. Keeping hens was positively associated with mothers' egg consumption; the egg consumption frequency was almost double among mothers who kept hens, compared to those who did not. In addition, the egg consumption

frequency of mothers who kept goats/sheep was 2.3 times higher than for mothers who did not keep these animals. This finding may explain the indirect positive association of goat/sheep husbandry with increasing the purchasing ability of the household (*Workicho et al., 2016*).

More than 95% of the mothers had consumed dairy products during the month prior to this study. In most rural areas, the watery portion of the milk after fat extraction is used. Otherwise, whole milk is rarely used to meet household demand. In this study, mothers who kept cows consumed dairy products 1.9 times more frequently than those who did not. A similar association was reported in studies performed in Ethiopia (*Daba et al., 2021*; *Melesse & Beyene, 2009*) and in the above mentioned study from Kenya (*Thumbi et al., 2015*). In general, our finding indicates that keeping cows is a positive predictor of frequent consumption of dairy products among rural households. Goat/sheep ownership was also positively associated with mothers' milk consumption frequency. This association can be explained in two ways: by purchasing milk from the local market after selling live goats/sheep, and by using goat's milk. There is a study reporting on the use of goat's milk in Ethiopia (*Azeze et al., 2015*).

The age of the child was associated significantly with all ASF consumption, whereby older children consume ASFs more frequently than younger children. This finding was in line with a study undertaken in four different regions of Ethiopia (*Potts, Mulugeta & Bazzano, 2019*). This association may also be due to cultural practices whereby some ASFs are forbidden for young children (*Haileselassie et al., 2020*). In addition to accessibility and affordability, poor digestibility of protein among children is also an important issue. As the digestive system of infants is immature, the tough structure of meat may cause lower utilization among young children. However, type and quality of protein, fat, and dietary fibre content, and the cooking methods need to be considered (*Cutroneo et al., 2023*; *Nicklaus & Tournier, 2023*). We also found that in larger families, children consumed dairy products less frequently. This finding shows that larger families are mostly unplanned and indicative of women and girls being uneducated and not employed (*Kim, 2016*). These larger families are poor and cannot afford most ASFs (*Haileselassie et al., 2020*). Hence, children and mothers in larger families do not consume, or less frequently consume, ASFs that entail relatively high costs (*Daba et al., 2021*). Furthermore, the educational background of the mother is found to have a significant positive association with most ASF consumptions among children and mothers (*Hamza et al., 2022*). This finding further supports the notion that education plays an important role in enhancing knowledge and awareness among mothers regarding appropriate child feeding practices and dietary diversity for optimal growth and overall well-being (*Dhami et al., 2021*; *Kim, 2016*).

## Strengths and limitations of our study

Our study had several strengths: it was a community-based study using a random sampling technique, collecting a comprehensive set of data at the child, mother and household level, and using the ordinal logistic regression model to describe associations between the outcome and exposure variables. Checking independent associations between each livestock type with each ASF is also an additional strength, since this helps to address

indirect associations between livestock ownership and consumption of specific ASFs. Moreover, assessing the food frequency over one month helps to address food items with infrequent consumption. On the other hand, the limitations of this study include the inherent limitations of the cross-sectional design: one can never rule out reverse causality. Between the mapping and the actual cross-sectional study there was replacement of households due to people leaving their residence prior to the data collection. Recall bias for self-reported data, such as age and food frequencies, could be a source of bias. To minimise the recall bias, however, we used the strategies described under data quality control measures in the method section. Survivor bias might also exist, since our samples were mother-child pairs, and mothers who had lost their children were not included. We did not assess the food portions consumed, the number of livestock and to whom they belonged (the mother or the father), which is also a limitation.

## CONCLUSION

In conclusion, meat consumption was generally low among children aged 6–24 months and their mothers in a rural district of Sidama region. However, dairy products and eggs were served more frequently, indicating a potential for increased ASFs consumption. Livestock ownership was identified as a significant positive factor for ASF consumption in rural settings. To promote ASF consumption in rural settings, it is recommended to encourage livestock diversification at household level. In addition, positive strategies might help to improve access to ASFs in rural households. Incorporating nutrition education into community health care services should be considered.

## ACKNOWLEDGEMENTS

We thank the mothers and caregivers with their young children for their participation in our study. We are grateful to the Sidama regional state health office and the Dale district health department for their cooperation and support during our data collection. We express our gratitude to the Dale and Wonsho Health and Demographic Surveillance Site for their cooperation to access the profile of Dale district. We are also thankful to data collectors and supervisors for the commitment they demonstrated during the data collection.

### Funding

This study was funded by the South Ethiopia Network of Universities in Public Health (SENUPH), which in turn was funded by the Norwegian Program for Capacity Development in Higher Education and Research for Development (NORHED), grant number: ETH-13-0025. The funders had no role in study design, data collection and analysis, decision to publish, or preparation of the manuscript.

## Grant Disclosures

The following grant information was disclosed by the authors:

South Ethiopia Network of Universities in Public Health (SENUPH).

Norwegian Program for Capacity Development in Higher Education and Research for Development (NORHED): ETH-13-0025.

## Competing Interests

Bernt Lindtjorn is an Academic Editor for PeerJ.

## Author Contributions

- Tsigereda Kebede conceived and designed the experiments, performed the experiments, analyzed the data, prepared figures and/or tables, authored or reviewed drafts of the article, and approved the final draft.
- Selamawit Mengesha Bilal conceived and designed the experiments, analyzed the data, authored or reviewed drafts of the article, and approved the final draft.
- Bernt Lindtjorn conceived and designed the experiments, analyzed the data, authored or reviewed drafts of the article, and approved the final draft.
- Ingunn M. S. Engebretsen conceived and designed the experiments, performed the experiments, analyzed the data, prepared figures and/or tables, authored or reviewed drafts of the article, and approved the final draft.

## Human Ethics

The following information was supplied relating to ethical approvals (*i.e.*, approving body and any reference numbers):

Hawassa University, College of Medicine and Health Sciences Institutional Review Board, Ethiopia and Norwegian Regional Ethical Committee (REK), Norway

## Data Availability

The raw data is available in the Supplemental File.

## Supplemental Information

Supplemental information for this article can be found online at http://dx.doi.org/10.7717/peerj.16518#supplemental-information.

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
