# Peer review of "Does livestock ownership predict animal-source food consumption frequency among children aged 6–24 months and their mothers in the rural Dale district, southern Ethiopia?"

_PeerJ, doi:10.7717/peerj.16518_

## Round 0.1 · original submission · Major Revisions

Please provide a comprehensively revised version addressing the editorial comments and a detailed rebuttal letter.


Reviewer 1 ·

Basic reporting

Check English throughout the paper

Introduction
1. Check consistency of second sentence (line 64-65)
2. Old reference e.g. Pensel 1998 (line 72)
3. Sharpen the gap. You describe the problem using studies done in Ethiopia but justify unavailability of evidence.

Experimental design

Methods
4. Update study setting e.g. number of districts in the region and cite
5. Study period is missing in both abstract and methods
6. Line 122 “… mainly farmers, …”
7. Sample size in original study (990) is slightly different from sample size submitted as supplemental file – Fig 1 (985).
8. Line 134 “… a child…” does this mean each HH had a child? Clearly state number of children or if it was pair of child-mother and was equal to #HH, state that.
9. Line 147 “… respectively”. Not clear
10. Line 149 “… less than once…” less than one is zero. Check.
11. Line 186: I think livestock appeared in wealth index and also as independent variable. Describe
12. How was the ownership measured? Did you consider number, of cows for example?
13. Ownership of cow doesn’t mean the HH have access to milk as cows may not give milk at that specific time, and the same is true about hen. Do you have data to rule out it? I mean whether the existing cow/hen was giving milk/egg or not.

Validity of the findings

Results
14. Line 236: the word respectively confuses. What is its use here?
15. Line 253: delete
16. Expression of numbers and percent is inconsistent. After describing size of group, for example male or female, reporting % or number (%) is enough.
17. Line 257: Similarly, meat consumption was 26.8% (111 out of 418 male children) and 25.6% (112 out of 433 female children) can be written as “Similarly, meat consumption among male children was 111 (26.8%) and 112 (25.6%) among females.”
18. Add crosstab of ownership of livestock and consumption of products
19. Table 3: check counts in cell and total


Discussion
20. Consumption is usually associated with income. How was association with wealth index?
21. How ownership of sheep/goat improve egg consumption? Do you think they sell sheep to buy eggs if they keeping assets?
22. The Ethiopian orthodox followers do not eat animal products during fasting. Was there fasting at the time of data collection or do you have religion data and assessed the association?
23. Did you include urban kebeles in the district in your sampling? If yes, discuss. Usually, urban settings don’t own livestock but consume products more.

Additional comments

Abstract
1. Make numbers consistent e.g. ownership of cows is in % while sheep in numbers
2. Start sentence with text. Fourth sentence of results (line 47)

·

Basic reporting

It needs improvement

Experimental design

OK

Validity of the findings

1. Check normal distribution of the sample size.
2. Some designs indicated in methodology not described in results
3. Discussion and conclusion needs major revision

Additional comments

Everything is stated in the manuscript. Answer line by line

---

## Round 0.2 · Minor Revisions

Please address the reviewers' comments and submit a revised version of the manuscript at your convenience.

Reviewer 1 ·

Basic reporting

My concerns are addressed.

Experimental design

My concerns are addressed.

Validity of the findings

1. Line 288: “For most rural households in Ethiopia, cows are not used as a source of meat, …”. I think this is general truth. I don’t know if there’s a community that uses household cow as source of meat (you too stated it in line 311). It is too much for household. Revise sentence.
2. Line 294-304: Point by point response statement about this issue is clearer than this paragraph. It is more likely to be associated with wealth status of households. Just a question: how the geographical and cultural differences contributed to association. Any point to describe/compare settings?

Additional comments

Check Table 2 (the display)

·

Basic reporting

OKay

Experimental design

OKay

Validity of the findings

Okay

Additional comments

after accepting the track change, some editorial issues should be addressed. accepted for publication

---

## Round 0.3 · accepted · Accept

I am pleased to inform you that your manuscript has been accepted after addressing the comments raised by the reviewers. Thank you for your efforts!